# Use of Zebrafish Embryo Assay to Evaluate Toxicity and Safety of Bioreactor-Grown Exopolysaccharides and Endopolysaccharides from European *Ganoderma applanatum* Mycelium for Future Aquaculture Applications

**DOI:** 10.3390/ijms22041675

**Published:** 2021-02-07

**Authors:** Wan Abd Al Qadr Imad Wan-Mohtar, Zul Ilham, Adi Ainurzaman Jamaludin, Neil Rowan

**Affiliations:** 1Functional Omics and Bioprocess Development Laboratory, Institute of Biological Sciences, Faculty of Science, Universiti Malaya, Kuala Lumpur 50603, Malaysia; qadyr@um.edu.my; 2Bioresources and Bioprocessing Research Group, Institute of Biological Sciences, Faculty of Science, Universiti Malaya, Kuala Lumpur 50603, Malaysia; ilham@um.edu.my; 3Bioscience Research Institute, Athlone Institute of Technology, Dublin Road, N37 WO89 Athlone, Westmeath, Ireland; 4Environmental Science and Management Program, Institute of Biological Sciences, Faculty of Science, Universiti Malaya, Kuala Lumpur 50603, Malaysia; adiainurzaman@um.edu.my; 5Empower Eco Innovation Hub, Lough Boora, Co., R35 DA50 Tullamore, Offaly, Ireland

**Keywords:** *Ganoderma applanatum*, exopolysaccharide, endopolysaccharide, zebrafish, toxicology

## Abstract

Natural mycelial exopolysaccharide (EPS) and endopolysaccharide (ENS) extracted from bioreactor-cultivated European *Ganoderma applanatum* mushrooms are of potential high commercial value for both food and adjacent biopharmaceutical industries. In order to evaluate their potential toxicity for aquaculture application, both EPS (0.01–10 mg/mL) and ENS (0.01–10 mg/mL) extracts were tested for Zebrafish Embryo Toxicity (ZFET); early development effects on Zebrafish Embryos (ZE) were also analyzed between 24 and 120 h post-fertilization (HPF). Both EPS and ENS are considered non-toxic with LC_50_ of 1.41 mg/mL and 0.87 mg/mL respectively. Both EPS and ENS did not delay hatching and teratogenic defect towards ZE with <1.0 mg/mL, respectively. No significant changes in the ZE heart rate were detected following treatment with the two compounds tested (EPS: 0.01–10 mg/mL: 176.44 ± 0.77 beats/min and ENS: 0.01–10 mg/mL: 148.44 ± 17.75 beats/min) compared to normal ZE (120–180 beats/min). These initial findings support future pre-clinical trials in adult fish models with view to safely using EPS and ENS as potential feed supplements for supplements for development of the aquaculture industry.

## 1. Introduction

Over the past two decades, medicinal mushroom exopolysaccharides (EPS) and endopolysaccharide (ENS) have been increasingly exploited for human benefits including clinical trials [1], neurological protection [2], potent in vitro biological activities [3], chicken patty [4], anti-oral cancer [5], and breast cancer models [6]; however, there is still limited development as yet on their commercial use by the aquaculture industry. Development and exploitation of medicinal mushroom products has significant potential to improve farmed fish health, where appropriate innovation in fish feed and formulation potentially offers safe, affordable and future intensive sustainability needs [7,8,9]. Recent evidence suggests the use of bioreactor-grown medicinal mushroom *Ganoderma* sp. extracts in aquaculture-like areas including fish-feed [10], fish toxicity [11], and have passed potential clinical application [12]. These potential applications are largely attributed to its bioactive EPS-ENS components with immuno-stimulatory properties, which can be scaled efficiently and repeatedly for commercial use by deploying a specifically-tailored mushroom bioreactor system [13,14].

Currently, there is an innovation gap in terms of specific use and development of fungal biotechnology for meeting emerging needs in the aquaculture industry globally. However, there are some examples of specific mushroom usage including Shiitake fruiting body for fingerlings of Carps [15] and Rainbow trout feed [16]; thus, use of Ganoderma- derived EPS-ENS that comprise beta-glucans warrants further investigation and testing. Mushroom scientists have previously reported that *G. applanatum* EPS-ENS demonstrated promise as novel disease mitigation therapeutic [17,18,19], suggesting potential for use as disease prevention counter-measures for adjacent aquaculture industry in the form of prebiotics [20]. 

The creative idea of using extracts from naturally-occurring algae to address key challenges and opportunities in fish health has been exemplified by Marino et al., [21] who studied use of algal extracts as potential alternatives to that of antibiotics for farming sea bass and gilthead sea bream. Marino et al., [21] highlighted increased interest in the use of natural-alternatives materials to that of frontline antibiotics and biocides, particularly where there is a trend towards producing farmed high-value fish with organic status. Cascio et al., [22] reported on the potential “game-changer” use of a microalgae (Spirulina) targeting beneficial effects in the gastrointestinal tract of Zebrafish; this has implications for potentially stimulating the breeding of animals. Such studies show the importance of Zebrafish (*Danio rerio*) assay due to the ability of Zebrafish stomach system to recognize small extracts such as mRNA and immune-priming genes. The use of beta-glucans-derived from medicinal mushrooms, yeast, algae, seaweed and so forth is of increasing interest as these bio-based therapeutics have shown potential for immune-priming of fish for disease mitigation and for added food security. However, the vast majority of studies to date have focused on extraction of bioactives from raw materials, but not from bioreactor grown systems that are important for larger pilot scale and commercial deployment. It is therefore necessary to test and verify the toxicity status of EPS-ENS polysaccharide extract prior to pilot and commercial use; such as, the use of a zebrafish larvae model to evaluate these bioactive substances as part of a fish feed formulation study [23,24]. For example, use of antibacterial polysaccharides from *Undaria pinnatifida* (macroalgae) was reported by Rizzo et al., [25] for the treatment of prominent aquaculture disease *Vibrio harveyi* causes death of marine fish [26] due to vasculitis, gastro-enteritis and eye lesions.

Our study used the ZFET strategy as it was previously reported to be a fast, affordable and sensitive approach [11]; it also offers greater flexibility in terms of using multiple sample insertions to evaluate toxicological effects due to early stages of embryonic development [11] and for continuity in terms of relevance to the gastrointestinal anatomy in small intestine mammals [22]. This study reports on the use of ZFET assay for testing European *G. applanatum* extracts using seven different EPS-ENS concentrations that specifically addresses LC_50_, embryonic hatching delays, teratogenic defect, and heart rate response with clear microscopic images.

## 2. Results

### 2.1. Effects of EPS and ENS on the Survival Rate of Zebrafish Embryos

In the ZFET assay, the survival rate of the embryos of the zebrafish in the range of 0 to 120 h old at the cell culture concentration of 0.01 to 10 mg/mL was analyzed. Based on standard, the normal hatching for zebrafish embryos is median of 48 to 72 h of post-fertilization (HPF), thus the survival rate of the embryos (prior hatch) and larva (post hatch) treated with EPS-ENS extract was determined for maximum of five days (120 h). In Figure 1, we observe that 100% of untreated embryos survived for up to 120 HPF. However, among pretreating cells with EPS-ENS doses of 0.01 to 1 mg/mL, the cell survival rate was 88% at 72 HPF. The survival rate dropped from 88% at 72 HPF to 0% for all cells pretreated at the highest EPS-ENS concentrations (5–10 mg/mL). No embryos at >72 HPF managed to survive and developed at EPS concentration >5 mg/mL (Figure 1a). In the meantime, Figure 1b, on low concentrations (0.5–1 mg/mL) of ENS extract, an average drop in survival rate was relatively small (85%), while at higher concentrations (5–10 mg/mL), a lower survival rate (<30%) was observed at just 4 days of age (96 HPF). No embryo survived after 120 HPF for ENS at concentrations >5 mg/mL. Collectively, it shows that EPS-ENS delay hatching optimum concentrations of less than <1 mg/mL.

### 2.2. Performance of EPS-ENS Doses on the Zebrafish Embryos Mortality

Overall, extracts of European *G. applanatum* EPS-ENS have lethal effects that vary with dose and time. In Figure 2, at low concentrations, EPS (<3 mg/mL) and ENS (<1 mg/mL) were both able to prevent 90% of zebrafish embryos from dying. However, at high concentrations, 4 mg/mL EPS and 2 mg/mL ENS did not perform better (with low survival rate) throughout the embryo development stage, showing dead embryos post 72 HPF. Consequently, the LC_50_ value of zebrafish embryos exposed to EPS extract tested was 1.41 mg/mL, whereas the LC_50_ value for ENS extract tested was much lower than this at 0.87 mg/mL.

### 2.3. Performance of EPS and ENS Doses on the Zebrafish Embryos Hatching Capability

Subsequently, different EPS-ENS concentrations of extract may affect the hatchability percentages. Figure 3 shows a trend of decreasing hatching rates with increased concentrations of the extract (>1 mg/mL) once treated into zebrafish embryos. It also depicts the hatching rate during (a) EPS treatments at 0.01–10 mg/mL and (b) ENS treatments at 0.01 to 10 mg/mL during 0 to 120 HPF period. For EPS, there was no visible changes in terms of hatching capability at concentrations less than 1 mg/mL, but the rate dropped to 35% at 48 HPF once exposed to 5 mg/mL. Zebrafish did not hatch or survive when treated with a concentration of 10 mg/mL EPS, indicating that high amounts of EPS causes strong suppression of zebrafish embryo at 24 HPF. On the other hand, less than 81% of the embryos hatched on 48 h treatment with ENS at concentrations >0.5 mg/mL; although, zebrafish larvae treated with ENS concentrations at 5 to 10 mg/mL showed the lower hatching rate (<65%) due to high mortality rate after 72 HPF.

### 2.4. The Effects of EPS and ENS on the Heart Rate of Zebrafish Embryos

In the development of many model organisms, zebrafish heart represents the core of functional organ [27]. Based on Figure 4, both extracts of EPS and ENS at low concentrations <1 mg/mL, showed no significant difference towards the heart rate of untreated zebrafish embryo at 96 HPF. The heart rate at day 4 (96 HPF) for EPS (Figure 4a) treatment was healthily recorded at 176.44 ± 0.77 beats min^−1^ at concentrations <1 mg/mL compared to normal average zebrafish embryonic heart rate (120–180 beats min^−1^). Meanwhile, ENS (Figure 4b) treatment also gave normal heart rate (148.44 ± 17.75 beats min^−1^) at concentrations <1 mg/mL. Since EPS extract at 5 and 10 mg/mL killed the zebrafish at 96 HPF, the embryonic heart rate at these concentrations were omitted. However, the embryo was barely alive when treated with 5 mg/mL of ENS (90 beats min^−1^) and unrecorded at 10 mg/mL due to death. Both EPS and ENS treatments verified the previous report in which the normal heart rate of zebrafish embryo nearly mimics human condition at 120 to 180 beats min^−1^ [28].

### 2.5. Performance of EPS and ENS Doses on the Morphology of Larvae and Zebrafish Embryos Development

The performance of EPS and ENS doses were observed from 0 to 120 HPF period particularly on larvae and zebrafish embryos morphological defects. Figure 5 depicts no visible teratogenic effect of either EPS or ENS at concentrations <1 mg/mL on the embryogenesis stages at 120 h after exposing the embryos to them. Based on these data, it clearly suggested that either EPS or ENS has affected larvae and zebrafish embryos development before and after hatching.

As both zebrafish embryo and larvae development were unaffected when treated with 1 mg/mL EPS and 1 mg/mL ENS, various abnormalities were observed as the concentration significantly spiked to 5 mg/mL EPS (Figure 6) and 1.31 mg/mL ENS (Figure 7). One of the most prone abnormalities was observed for ENS showing the period of coagulated embryos from 24 HPF (segmentation) to 48 HPF (pharyngula), and loss of yolk sac, resulting in unhatched embryos even after 120 HPF. Meanwhile, ENS-treated zebrafish managed to hatch at 72 HPF, however with tail malformation and broken blood cells after 120 HPF. Most of normal features were also absent such as fin, gut, and melanophores.

## 3. Discussion

In this study, both EPS and ENS extracted from bioreactor-grown European *G. applanatum* were analyzed for acute toxic effects in treated zebrafish embryo. EPS-ENS originates from edible mushroom species and has potential to be used as high-value product in the form of dietary fish feed supplement [29]. Recently, use of EPS-ENS (or beta-glucan) extracts has increased in interest for various applications, such as for preventing biofilms [30] and for fish-feed [10]. Beta-glucans derived from mushrooms and other sources, have also been reported as potential new therapeutic to help fight Coronavirus disease (COVID-19) in the form of potential immunotherapies [31]; to alleviate use of frontline antibiotics for combatting resilient bacteria [32]; as an alternative to the use of silver nanoparticles [33]; or for other potential applications such as cosmetics [34], immune priming or boosting [12], and prevention of cell death [2]. However, study on the toxicity of these specific compounds are lacking, including use of Zebrafish testing, which would help inform product development and applications. 

Using this ZFET technique, healthy zebrafish embryos, or larvae, have been used as surrogate animal model, which offers several advantages such as efficient, fast, reliable and comparable to human model [23,35]. This suits the “Zebrafish 3.0” toxicity model as the strategy share many physiological and cellular characteristics with higher vertebrates. In addition, the current ZFET assay uses 96-well plate that have a direct contact with EPS-ENS, mimicking the close contact with the demersal characteristic of zebrafish embryo. Secondly, extra-uterine and transparency can be examined, representing clear phenotypic embryonic development changes. In this study, teratogenic and embryotoxic effects evaluations are essential for EPS-ENS to determine the concentration that is safe for consumption. With booming global health industry and products from plants and fungi, claimed to have strong pharmacological responses, most of them are still dubious on the toxicological profiles.

In this study, fertilized zebrafish embryos were exposed to concentrations of *G. applanatum* extract, EPS (0.01–10 mg/mL) and ENS (0.01–10 mg/mL) that were shown to be non-toxic using this ZFET technique. Overall, EPS at concentrations <1 mg/mL and ENS at <1 mg/mL did not cause delay hatching towards the embryo and with 88% survival rate at 24 to 120 HPF. Additionally, there were no significant changes in both EPS and ENS at concentration <1 mg/mL on the embryo heart rate compared to the normal ones. Moreover, teratogenic effect and zebrafish embryo abnormalities can only be observed at concentrations >5 mg/mL and >1 mg/mL in EPS and ENS, respectively. The test showed that EPS has a higher LC_50_ value of 1.41 mg/mL, meaning that it is better than ENS with lower LC_50_ value (0.87 mg/mL). Even though both EPS and ENS extracts originated from *G. applanatum* mycelium, they may differ in terms of compound composition which originated from fruiting body and different mycelial extraction procedures [36,37,38]. Former research verified that similar EPS from the sister *G. lucidum* mycelium exhibit a broad range of bioactivities, including immunostimulant and antitumorigenic effects [39], which are higher than those of the fruiting bodies [34]. Meanwhile, ENS has given a lower LC_50_ value than EPS due to its different mycelial extraction methodology; EPS is directly extracted from the surface of fungal mycelium while ENS has undergone series of physico-chemical extractions from the internal part of dried fungal mycelium [5].

Morphological deformities, including tail malformation, could restrict the embryo’s ability to break the chorion (Ch in Figure 6) and hatch out post 5 days. Likewise, the absence of heartbeat and coagulated embryo is considered as lethal effects. The work of Dulay et al. [40] has confirmed that tail malformation in zebrafish embryo does occur when exposed to 10 mg/mL of *Ganoderma* sp. fruiting body extract. In addition, it depicted that the safety threshold concentration is <10 mg/mL for any Ganoderma sp. fruiting body extract [40], which is in-line with the current data described for *G. applanatum* EPS and ENS possessing less toxic responses.

Some medicinal mushrooms have also been studied on the effects of toxicity towards zebrafish embryos in comparison to *Ganoderma* species. Exposure of termite-mound mushroom *Termitomyces clypeatus* extract at concentrations of >0.1% reduces the zebrafish embryos hatchability and <50% extract gave visible teratogenic effects post 48 HPF [41,42]. On the other hand, exposure of grey oyster mushroom *Pleurotus ostreatus* ethanol extract killed the zebrafish embryos using 2.5 and 5% concentrations post 12 HPF, while obstructed growth and tail malformation can be observe at 1% dose [43].

Recent studies have reported that *G. applanatum* polysaccharides are heavily used as food supplements for human species [19,44,45] to improve growth and immunity but none was tested as aquatic feed. The closest comparison were only by its popular sister *G. lucidum* at acceptable concentrations ranging from 1.0 to 1.5 g/kg on giant freshwater prawn (*Macrobrachium rosenbergii*) [46] and grass carp (*Ctenopharyngodon Idella*) [47], which enhance innate immune response and development. Therefore, to gain safety concentration for both aquaculture feed and human drug, both EPS and ENS extracts from *G. applanatum* toxicity reports are essential. Henceforth, the current novel ZFET data may provide useful information for assessing the potential health risks of the EPS-ENS consortia. Still, further tests are merited to evaluate the LC_50_ value of EPS-ENS extract for informing large-scale human trials and for larger animals (e.g., pig, rabbit, and adult trout), before this innovation can be deployed for commercial use.

This constitutes the first toxicology study that addresses bioreactor-cultured European *G. applanatum* (temperature climate mushroom) extracts where findings are compared to that of extracts derived from its sister *G. lucidum* (Table 1). Three studies have previously reported on the evaluation of EPS from *G. lucidum* using ZFET assay (2648 μg/mL) [11], normal prostate cell line (500 μg/mL) [48], and normal human lung cell (1000 μg/mL) [49]. The findings from this present study reports on testing and evaluation of both non-toxic mycelial EPS (1410 μg/mL) and ENS (870 μg/mL) of stirred-tank bioreactor grown *G. applanatum* samples using the zebrafish 3.0 toxicity model. Moreover, this study provided evidence to support the safe use of European *G. applanatum* bioactive polysaccharides via Zebrafish model.

## 4. Materials and Methods

### 4.1. European Ganoderma Applanatum Bioreactor Fermentation

European *G. applanatum* strain BGS6Ap was isolated by Prof. Dr. Anita Klaus from the Republic of Serbia in the wild region of Mount Kosmaj at temperate environment (18 °C–24 °C), coordinate of 44°27′57″ N, 20°33′52″ E, and maintained at 4 °C on Malt Extract Agar (MEA) slants prior to fermentation setup. The bioreactor fermentation of *G. applanatum* consists of double-seed culture stages. The seed culture medium condition was previously optimized for Ganoderma sp. liquid batch fermentation at fixed metrics (g/L): Yeast extract 1, Glucose 30, KH_2_PO_4_ 0.5, MgSO_4_ 0.5, and NH_4_Cl_4_ 4 [50]. The fungus was cultivated for ten days at pre-optimized condition of 25 °C, pH 6, 10 g/L of glucose and 150 rpm in a 5-L stirred-tank (STR- Labfors, Infors H-T, Basel, Switzerland) bioreactor [51] to produce fungal biomass. 

### 4.2. G. applanatum EPS-ENS Preparations

The fungal biomass from the harvested *G. applanatum* culture was subjected to Buchner funnel filtration, followed by triple washing with distilled water. To obtain EPS, the filtrate was precipitated by the addition of four volumes of 99% (*v*/*v*) cold-ethanol-shock treatment and left overnight at 4 °C. After 24 h, crude EPS was formed naturally by agglomeration, and centrifuged at 10,000 rpm for 20 min. Meanwhile, to obtain ENS, the initial washed-fungal biomass was dried in a dryer until it powdered. The dried biomass was mixed with 1 g:20 mL distilled water, then subjected to heat-extraction (24 °C for 30 min) [52]. The resulting crude ENS was precipitated using the similar procedure of crude EPS. Finally, both EPS and ENS crude extracts were slow-dried to constant weight prior to toxicity studies. Stock powder of dried EPS-ENS was diluted in embryo medium (Danio-SprintM solution) producing working solutions (10 mg/mL). The mixture was diluted in 2-fold serial dilutions obtaining six concentrations ranging from 0.01–5 mg/mL in a 96-well microplate (200 µL). Embryos as in embryo medium were assigned as standard control (untreated).

### 4.3. Zebrafish Maintenance and Breeding

The use of zebrafish was approved under the by the Institutional Animal Care and Use Committee (IACUC), Faculty of Biotechnology and Biomolecular Sciences, Universiti Putra Malaysia. Briefly, zebrafish of both sexes were placed into a breeding tank on the day the breeding tank was prepared. Following the first day, embryos were collected, washed, and then incubated in Danio-SprintM solution. Dead and coagulated embryos were discarded in an attempt to find healthy embryos [11].

### 4.4. Zebrafish Embryo Toxicity (ZFET) Assay

The standard of Organization for Economic Cooperation and Development (OECD) standard was used in the ZFET assay [53]. Initially, one zebrafish embryo per well system at 0 HPF were subjected to diluted EPS-ENS samples (200 μL) in 96-well microplates (Corning^®^ 96 Well Clear Polystyrene Microplate, Corning, NY, USA) at seven different concentrations ranging from 0.01–10 mg/mL. The EPS-ENS and untreated samples were tested with a total of 24 replicate of embryos per exposure group slight above minimum standard [54,55]. Successful treated embryos were incubated at room temperature (25–28 °C) for 5 days. The cumulative mortality and developmental malformations of embryos and larvae were observed and determined every 24 h from 0–120 HPF as previously suggested by Parenti et al., [56]. The results indicate that mortality, hatching rate, heart rate, morphological malformation or teratogenic defects were observed using THUNDER Imager 3D Live Cell & 3D Cell Culture & 3D Assay, Leica Microsystems GmbH, Wetzlar, Germany. Heartbeat (1 min) and lethal endpoints (no heartbeat and clear coagulation) were assessed as according to Taufek et al., [11]. Anomalous development includes yolk sac oedema, curved body, non-hatched, bent tail and pericardial oedema. The principle of toxicity (LC_50_) values for EPS-ENS derivative >1 mg/mL are considered relatively harmless, 0.1–1 mg/mL are considered practically non-toxic, 0.01–0.1 mg/mL are considered slightly toxic, 0.001–0.01 mg/mL are considered moderately toxic, 0.0001–0.001 mg/mL are considered highly toxic and <0.0001 mg/mL are considered super toxic. Appendix A: EPS-ENS Untreated vs. 0.01, 0.5 and 1 mg/mL treatments was supplied as Appendix A.

### 4.5. Statistical Analysis

ZFET response graphs and the lethal concentration at 50% (LC_50_) of treated samples toward zebrafish embryos were produced by using GraphPad Prism version 9.0 (GraphPad Software, Inc., 2365 Northside Dr. Suite 560, San Diego, CA, USA). Heart rate response was presented as mean ± standard error of mean (S.E.M) from three different animals. One-way analysis of variance (ANOVA) was used to carry out the significant differences with a post hoc test via Dunnett’s Multiple Comparison. The significant difference was considered at * *p* < 0.05, ** *p* < 0.01 and *** *p* < 0.001 between the means of treated group as compared to embryos in embryo medium.

## 5. Conclusions

In conclusion, this represents the first study to report on the use of zebrafish embryo toxicity (ZFET) assay on bioreactor-grown European medicinal mushroom *G. applanatum* EPS and ENS extracts. EPS (LC_50_: 1.41 mg/mL) was harmless while ENS (LC_50_: 0.87 mg/mL) are practically non-toxic. The ZFET assay offers a fast, affordable, robust, and efficient early development approach to evaluating extracts from medicinal fungi for future use in aquaculture [57]. Findings will inform and guide future “foodomics” research spanning molecular toxicology and linked proteomics and metablomics as it relates to targeting specific transcripts for biosensor development that will support end-to-end monitoring of value chain, including using of machine learning for mitigating food waste and improving “circularlity” for the food sector. Specifically, this study provides new knowledge that will inform efficacy for future immune-priming and disease mitigation innovation for global food security.

## Figures and Tables

**Figure 1 ijms-22-01675-f001:**
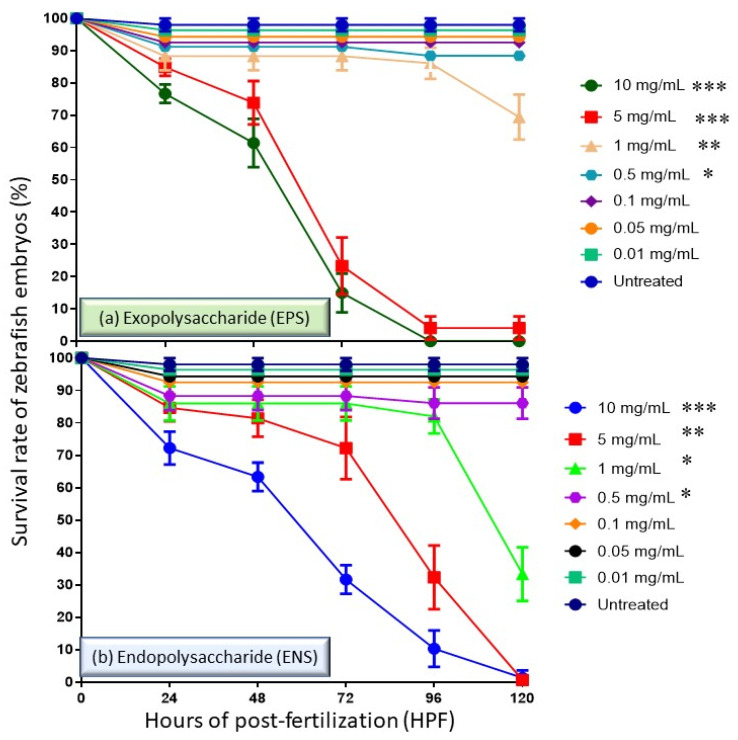
The performance of *Ganoderma applanatum* (**a**) EPS and (**b**) ENS extract at concentrations of 0.01–10 mg/mL on the survival rate of zebrafish embryos at 0–120 h. No embryos survived for both samples at concentration tested > 5.0 mg/mL after 96 h post fertilization (HPF). Symbols: * *p* ˂ 0.05, ** *p* < 0.01 and *** *p* < 0.001.

**Figure 2 ijms-22-01675-f002:**
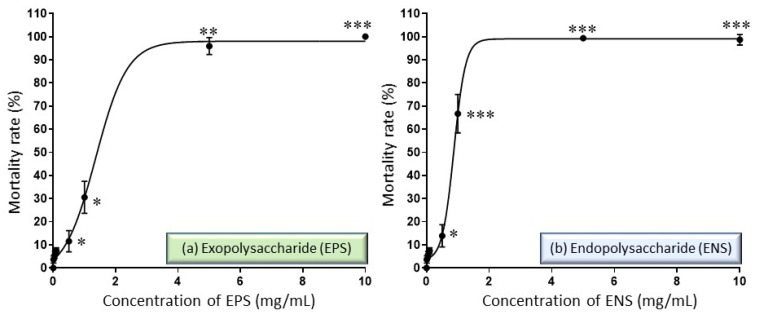
Effect of *Ganoderma applanatum* (**a**) EPS extract at concentrations of 0.01–10 mg/mL and (**b**) ENS at concentrations of 0.01–10 mg/mL on zebrafish embryos mortality rate after 120 HPF. The LC_50_ value for EPS extract was 1.41 mg/mL while LC_50_ value for EPS extract was 0.87 mg/mL. Symbols: * *p* ˂ 0.05, ** *p* < 0.01 and *** *p* < 0.001.

**Figure 3 ijms-22-01675-f003:**
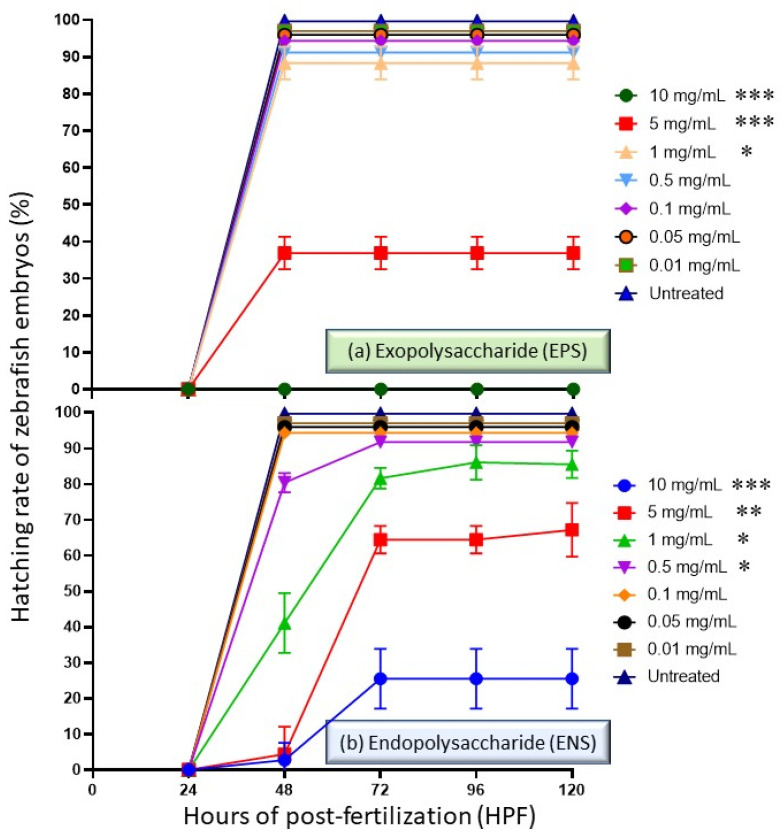
Hatching rate of zebrafish embryos at 0 to 120 h of post exposure with *Ganoderma applanatum* EPS and ENS extract at concentrations of 0.01–10 mg/mL. (**a**) For EPS, low hatching rate (<40%) was observed at concentration 5.0 mg/mL due to high mortality rate. Meanwhile, (**b**) for ENS, low hatching rate (<30%) was observed at concentrations 10 mg/mL due to high mortality rate. High hatching rate was observed at concentrations >1.0 mg/mL (>80%). Symbols: * *p* ˂ 0.05, ** *p* < 0.01 and *** *p* < 0.001.

**Figure 4 ijms-22-01675-f004:**
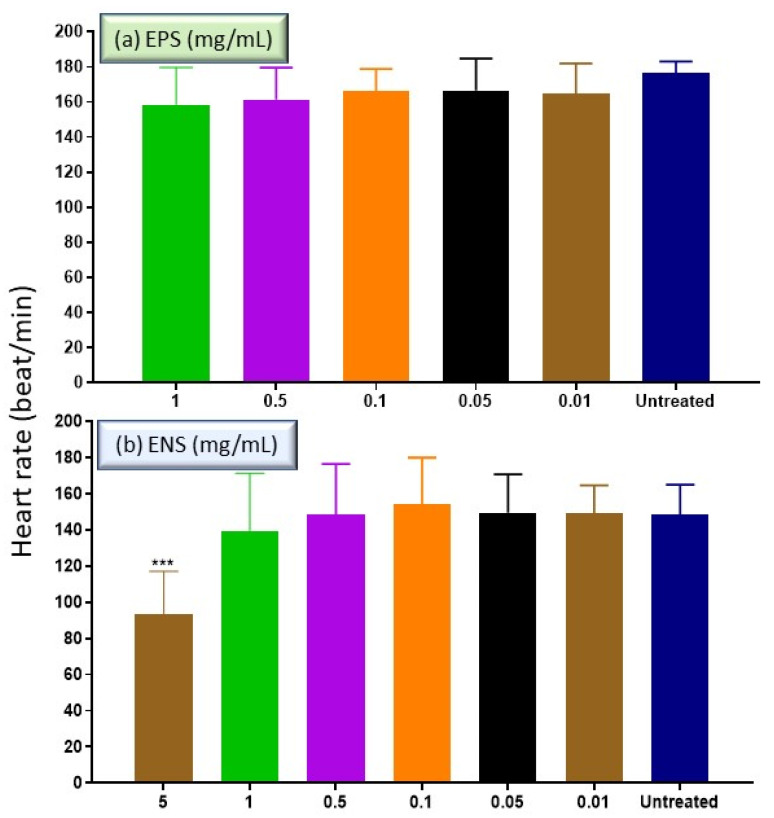
Effect of *Ganoderma applanatum* EPS and ENS extract at concentrations of 0.01–1.0 mg/mL on heart rate of zebrafish embryos at 96 HPF. For (**a**) EPS, no data at concentrations >5.0 mg/mL due to embryo death. Meanwhile, (**b**) for ENS, no data at concentrations 10.0 mg/mL recorded due to embryo death. **** p* < 0.05 significantly different from the untreated group (zebrafish embryos in medium only).

**Figure 5 ijms-22-01675-f005:**
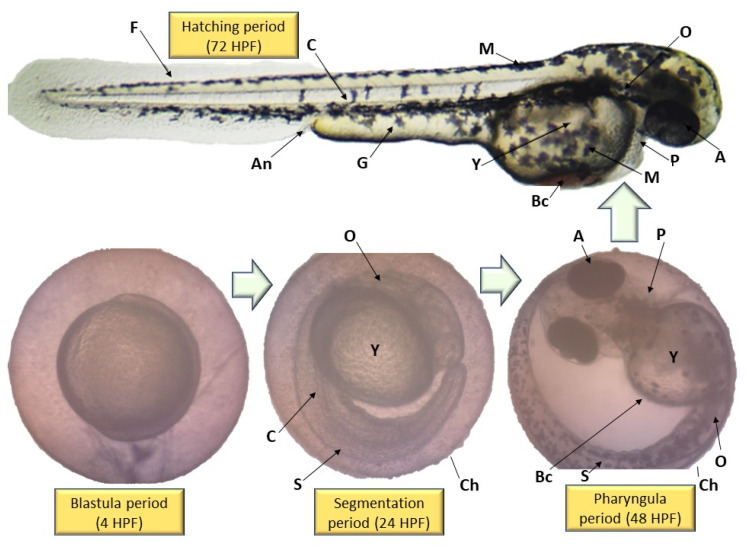
Effect of EPS-ENS extracts (0.01–1 mg/mL) of European *Ganoderma applanatum* showing normal zebrafish embryogenesis at different HPF development. The inverted microscope was used to produce the images and edited using paint 3D [11]. There were 4 periods depicted: Blastula (4 HPF), Segmentation (24 HPF), Pharyngula (48 HPF), and Hatching (72 HPF). Y—yolk sac; S—somites; P—pericardium; O—ear bud; M—melanophores; G—gut; F—fin; Ch—chorion; C—chorda; Bc—blood cells; An—anus; A—eye anlage. Scale bar = 0.5 mm.

**Figure 6 ijms-22-01675-f006:**
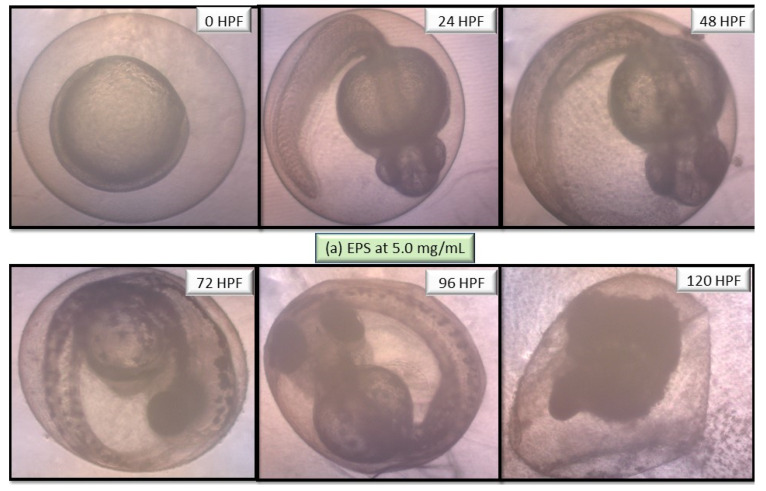
Images of zebrafish embryo and larvae development at 0 HPF, 24 HPF, 48 HPF, 72 HPF, 96 HPF and 120 HPF after treated with European *Ganoderma applanatum* at high EPS concentration of 5.0 mg/mL. Images were captured using inverted microscope at 100× (0 and 24 HPF) and 40× magnification (48–20 HPF).

**Figure 7 ijms-22-01675-f007:**
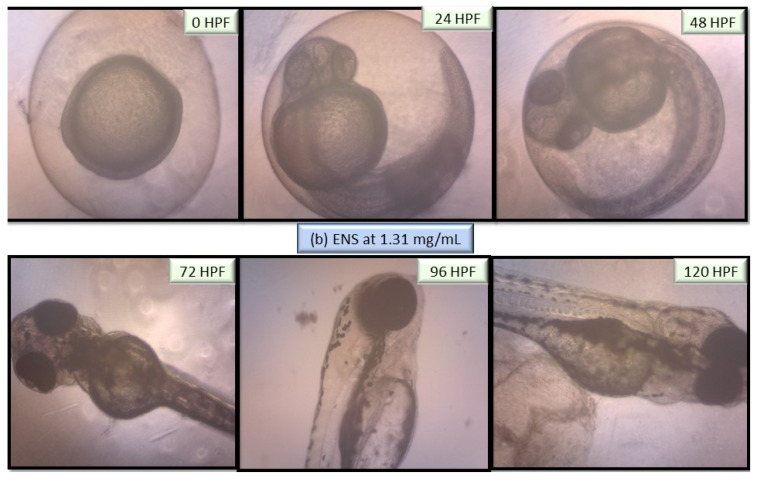
Images of zebrafish embryo and larvae development at 0 HPF, 24 HPF, 48 HPF, 72 HPF, 96 HPF and 120 HPF after treated with European *Ganoderma applanatum* at high ENS concentration of 1.31 mg/mL. Images were captured using inverted microscope at 100× (0 and 24 HPF) and 40× magnification (48–20 HPF).

**Table 1 ijms-22-01675-t001:** Comparison with published non-toxicity assessment of exopolysaccharide (EPS) and endopolysaccharide (ENS) from the mushroom *Ganoderma* sp.

Source	Toxicological Model	Non-Toxic Concentrations (μg/mL)		References
		Exopolysaccharide (EPS)	Endopolysaccharide (ENS)	
*G. applanatum* BGS6Ap	In vivo—Zebrafish embryos and larvae	1410	870	Current study
*G. lucidum* QRS 5120	In vivo—Zebrafish embryos and larvae	2648	NA	[11]
*G. lucidum* BCCM 31549	In vitro—normal human prostate cell line (PN2TA)	500	NA	[48]
*G. lucidum*	In vitro—normal human lung cell (WRL68)	1000	NA	[49]

## Data Availability

The data presented in this study are available on request from the corresponding author. The data are not publicly available due to privacy.

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
