# Peer review of "Use of Zebrafish Embryo Assay to Evaluate Toxicity and Safety of Bioreactor-Grown Exopolysaccharides and Endopolysaccharides from European Ganoderma applanatum Mycelium for Future Aquaculture Applications"

_ijms, 2021, doi:10.3390/ijms22041675_

Round 1

Reviewer 1 Report

The zebrafish is a widely used vertebrate model organism for the disease and phenotype-based drug discovery. Especially, zebrafish embryos are used for the rapid evaluation of toxicity of chemical compounds. This study assessed the toxicity of EPS-ENS extracts at seven different concentrations, on the survival rate, the hatching rate, the heart rate and morphology of zebrafish embryos and on larvae development. Overall the manuscript is interesting, but need a major revision before being accepted. The abstract is really confusing and bad written. The English must be improved. There are several mistakes, wrong verbal tenses and misleading periods

The aim of the study is not clear, why are you talking about red Hybrid Tilapia if the study is focused on ZFET. You should rewrite the abstract according to this and focus your discussion on zebrafish toxicity. Moreover, it cannot be concluded that these compounds can be safely applied as feed substances in the human and aquaculture industry based on a simple ZFET. Perhaps you can conclude that could be interest to deeper test these compounds through pre-clinical trials in adult fish models.

Line 19 page 1: Tilapia is not a fungivore. It feeds mainly on phytoplankton or benthic algae. Additionally, insect larvae are of some importance, as are aufwuchs and detritus; juveniles tend to be more omnivorous than adults. Moreover, the species of red Hybrid Tilapia is (Oreochromis mossambicus × O. niloticus).

Lines 28-29. “suggesting that these can be safely applied as feed substances in the human and aquaculture industry.” I suggest to change this sentence into “as feed substances in the aquaculture industry” in accordance with the title of the Manuscript and with Discussions. As the genetic and physiologic makeup of zebrafish is comparable to humans, toxicities of chemical compounds are similar between zebrafish and humans. However, zebrafish is a valuable tool only in the early phase of drug discovery for the evaluation of toxicity and safety of the chemical compounds and additional studies are therefore additional studies are needed to evaluate if EPS/ENS could be safely employed in human feed industry.

Introduction is really short and does not well describe the state of art of the topic. You should improve it and better describe research endpoints regarding natural compounds application on fish, particularly on zebrafish and importance of ZEFT.

I would suggest some references to be possibly used and cited:

- Marino, F., Di Caro, G., Gugliandolo, C., Spano, A., Faggio, C., Genovese, G., ... & Santulli, A. (2016). Preliminary study on the in vitro and in vivo effects of Asparagopsis taxiformis bioactive phycoderivates on teleosts. Frontiers in physiology, 7, 459.

- Lo Cascio, P., Calabrò, C., Bertuccio, C., Iaria, C., Marino, F., & Denaro, M. G. (2018). Immunohistochemical characterization of PepT1 and ghrelin in gastrointestinal tract of zebrafish: effects of Spirulina vegetarian diet on the neuroendocrine system cells after alimentary stress. Frontiers in physiology, 9, 614.

- Rizzo, C., Genovese, G., Morabito, M., Faggio, C., Pagano, M., Spanò, A., ... & Gugliandolo, C. (2017). Potential antibacterial activity of marine macroalgae against pathogens relevant for aquaculture and human health. J. Pure Appl. Microbiol, 11(4), 1695-1706.

Results are well reported

Line 85 pag 3: change media with medium throughout the test

Line 165 pag 7: change pericard with pericardium

Line 168 pag7: ENS is missing after 1mg/mL

Line  220 – 234    “Although both extracts (EPS and ENS) are …………………for G. applanatum EPS and ENS.” You write that Ganoderma sp. EPS and Ens extracts are less toxic from which of G. applanatum; but I did not understand if this difference depends on the species of the mushroom or on the different extraction parts. Please clarified.

Lines 314-315. “The EPS-ENS and untreated samples were tested with a total of 12 replicate of embryos per exposure group.” For the evaluation of toxicity, each concentration of a compound needs a minimum of 20 embryos (Aspatwar et al., 2019; Gourmelon et al., 2016).

Aspatwar A, Hammaren MM, Parikka M, Parikka S. rapid Evaluation of Toxicity of Chemical Compounds using Zebrafish Embryos. 2019. Medicine. doi:10.3791/59315.

Gourmelon A, Delrue N. Validation in Support of Internationally Harmonised OECD Test Guidelines for Assessing the Safety of Chemicals. Adv Exp Med Biol. 2016; 856:9-32. doi: 10.1007/978-3-319-33826-2_2

In Figure 3, the number 100 is deleted in the ordinate axis

Author Response

REVIEWER 1

The zebrafish is a widely used vertebrate model organism for the disease and phenotype-based drug discovery. Especially, zebrafish embryos are used for the rapid evaluation of toxicity of chemical compounds. This study assessed the toxicity of EPS-ENS extracts at seven different concentrations, on the survival rate, the hatching rate, the heart rate and morphology of zebrafish embryos and on larvae development. Overall the manuscript is interesting, but need a major revision before being accepted. The abstract is really confusing and bad written. The English must be improved. There are several mistakes, wrong verbal tenses and misleading periods

Reviewer Comments

Rebuttal / Changes / Amendments

The aim of the study is not clear, why are you talking about red Hybrid Tilapia if the study is focused on ZFET. You should rewrite the abstract according to this and focus your discussion on zebrafish toxicity. Moreover, it cannot be concluded that these compounds can be safely applied as feed substances in the human and aquaculture industry based on a simple ZFET. Perhaps you can conclude that could be interest to deeper test these compounds through pre-clinical trials in adult fish models.

Thank you for your expert suggestions. The team has corrected the theme as EPS-ENS for pre-clinical trials in adult fish models

New abstract:

Natural mycelial exopolysaccharide (EPS) and endopolysaccharide (ENS) extracted from biore-actor-cultivated European Ganoderma applanatum mushrooms are of potential high commercial value for both food and adjacent biopharmaceutical industries. In order to evaluate their potential toxicity for aquaculture application, both EPS (0.01-10 mg/mL) and ENS (0.01-10 mg/mL) extracts were tested for Zebrafish Embryo Toxicity (ZFET); early development effects on Zebrafish Embryos (ZE) were also analysed between 24 and 120 h post-fertilization (HPF). Both EPS and ENS are considered non-toxic with LC50 of 1.41 mg/mL and 0.87 mg/mL respectively. Both EPS and ENS did not delay hatching and teratogenic defect towards ZE with <1.0 mg/mL, respectively. No significant changes in the ZE heart rate were detected following treatment with the two com-pounds tested (EPS: 0.01-10 mg/mL: 176.44±0.77 beats/min and ENS: 0.01-10 mg/mL: 148.44±17.75 beats/min) compared to normal ZE (120-180 beats/min). These initial findings support future pre-clinical trials in adult fish models with view to safely using EPS and ENS as potential feed supplements for supplements for development of the aquaculture industry.

Line 19 page 1: Tilapia is not a fungivore. It feeds mainly on phytoplankton or benthic algae. Additionally, insect larvae are of some importance, as are aufwuchs and detritus; juveniles tend to be more omnivorous than adults. Moreover, the species of red Hybrid Tilapia is (Oreochromis mossambicus × O. niloticus).

We have deleted this statement in Line 19

Lines 28-29. “suggesting that these can be safely applied as feed substances in the human and aquaculture industry.” I suggest to change this sentence into “as feed substances in the aquaculture industry” in accordance with the title of the Manuscript and with Discussions. As the genetic and physiologic makeup of zebrafish is comparable to humans, toxicities of chemical compounds are similar between zebrafish and humans. However, zebrafish is a valuable tool only in the early phase of drug discovery for the evaluation of toxicity and safety of the chemical compounds and additional studies are therefore additional studies are needed to evaluate if EPS/ENS could be safely employed in human feed industry.

We have amended the Lines 28-19 accordingly.

Together, both natural compounds EPS and ENS can be considered non-toxic, suggesting that they can be safely used as feed substances in the eco-sustainable aquaculture industry and pre-clinical trials in adult fish models

Introduction is really short and does not well describe the state of art of the topic. You should improve it and better describe research endpoints regarding natural compounds application on fish, particularly on zebrafish and importance of ZEFT.

Thank you for fantastic references and we have added a new paragraph reflecting the application of natural compounds on fish and importance of Zebrafish.

Line 55-75

Such ‘’microorganism extracts for fish’’ idea has been verified by Marino et al., [21] on the use of algal extracts for sea bass and gilthead sea bream antibiotics. Another study by Cascio et al., [22] provided a breakthrough knowledge by introducing a microalgae (Spirulina) on the gastrointestinal tract of Zebrafish that stimulates the breeding of animals. Such studies show the importance of Zebrafish (Danio rerio) assay due to the ability of Zebrafish stomach system to recognise small extracts such as mRNA and immune genes. It is therefore necessary to test and verify the toxicity status of EPS-ENS polysaccharide extract prior to pilot and commercial use, such as the use of a zebrafish larvae model to evaluate these bioactive substances as part of a fish feed formulation study[23] [24]. The use of antibacterial polysaccharides from Undaria pinnatifida (macroalgae) was verified byRizzo et al., [25] using antibacterial polysaccharides from Undaria pinnatifida (macroalgae) for the treatment of prominent aquaculture disease Vibrio harveyi, which caused the death of severe marine fish [26] due to vasculitis, gastro-enteritis and eye lesions.

Our study used the ZFET strategy as it was reported to be a fast, affordable, with multiple sample insertions and are more sensitive to toxicological effects due to early stages of embryonic development [11] and possessing similar gastrointestinal anatomy to small intestine mammals [22]. This study reports on the use of ZFET assay for testing European G. applanatum extracts using seven different EPS-ENS concentrations that specifically addresses LC50, embryonic hatching delays, teratogenic defect, and heart rate response with clear microscopic images.

Application on fish was cited

  • Marino, F., Di Caro, G., Gugliandolo, C., Spano, A., Faggio, C., Genovese, G., ... & Santulli, A. (2016). Preliminary study on the in vitro and in vivo effects of Asparagopsis taxiformis bioactive phycoderivates on teleosts. Frontiers in physiology, 7, 459.

Natural compounds (polysaccharides) on zebrafish

  • Rizzo, C., Genovese, G., Morabito, M., Faggio, C., Pagano, M., Spanò, A., ... & Gugliandolo, C. (2017). Potential antibacterial activity of marine macroalgae against pathogens relevant for aquaculture and human health. J. Pure Appl. Microbiol, 11(4), 1695-1706.
  • Zhang, X. H., He, X., & Austin, B. (2020). Vibrio harveyi: a serious pathogen of fish and invertebrates in mariculture. Marine Life Science & Technology, 1-15.

Importance of ZEFT

  • Lo Cascio, P., Calabrò, C., Bertuccio, C., Iaria, C., Marino, F., & Denaro, M. G. (2018). Immunohistochemical characterization of PepT1 and ghrelin in gastrointestinal tract of zebrafish: effects of Spirulina vegetarian diet on the neuroendocrine system cells after alimentary stress. Frontiers in physiology, 9, 614.

Results are well reported

Thank you and we have double checked the data again.

Line 85 pag 3: change media with medium throughout the test

We have changed media to medium throughout the test

Line 165 pag 7: change pericard with pericardium

Pericard has been changed to Pericardium

Line 168 pag7: ENS is missing after 1mg/mL

Typo error,

As both zebrafish embryo and larvae development were unaffected when treated with 1 mg/mL EPS and 1 mg/mL ENS

Line  220 – 234    “Although both extracts (EPS and ENS) are …………………for G. applanatum EPS and ENS.” You write that Ganoderma sp. EPS and Ens extracts are less toxic from which of G. applanatum; but I did not understand if this difference depends on the species of the mushroom or on the different extraction parts. Please clarified.

The difference depends on both on;

  1. Mycelium or fruiting body parts
  2. Different mycelium extraction procedures

The explanation has been corrected as below;

Even though both EPS and ENS extracts originated from G. applanatum mycelium, they may differ in terms of compound composition which originated from fruiting body and different mycelial extraction procedures [36-38]. Former research verified that similar EPS from the sister G. lucidum mycelium exhibit a broad range of bioactivities, including immunostimulant and antitumorigenic effects [39], which are higher than those of the fruiting bodies [34]. Meanwhile, ENS has given a lower LC50 value than EPS due to its different mycelial extraction methodology; EPS is directly extracted from the surface of fungal mycelium while ENS has undergone series of physico-chemical extractions from the internal part of dried fungal mycelium [5].

Lines 314-315. “The EPS-ENS and untreated samples were tested with a total of 12 replicate of embryos per exposure group.” For the evaluation of toxicity, each concentration of a compound needs a minimum of 20 embryos (Aspatwar et al., 2019; Gourmelon et al., 2016).

Aspatwar A, Hammaren MM, Parikka M, Parikka S. rapid Evaluation of Toxicity of Chemical Compounds using Zebrafish Embryos. 2019. Medicine. doi:10.3791/59315.

Gourmelon A, Delrue N. Validation in Support of Internationally Harmonised OECD Test Guidelines for Assessing the Safety of Chemicals. Adv Exp Med Biol. 2016; 856:9-32. doi: 10.1007/978-3-319-33826-2_2

The statement was corrected from our data analysis (12x2=24)

The EPS-ENS and untreated samples were tested with a total of 24 replicate of embryos per exposure group slight above minimum standard [54, 55]

In Figure 3, the number 100 is deleted in the ordinate axis

Technical error; Figure 1 and 3 has been corrected with the presence of 100

Reviewer 2 Report

The manuscript presents an interesting and useful set of experimental data on the effects of exopolysaccharides and endopolysaccharides on zebrafish embryos. However, the data on survival rate (Fig.1 ) and hatching rate (Fig. 3) of zebrafish embryos require further statistical analysis. It does not follow from the text whether the authors have adjusted the p-values for multiple comparisons. Each set of data represents 7 pairwise tests for the difference between each treatment with control. All the p-values should be corrected using Bonferroni corrected, i.e. each p-value should be multiple on 7. When it is done, the p-values for the groups exposed to low concentration of the tested chemicals should be non-significant.

Author Response

Point-by-point disposition to reviewer comments – Manuscript ID IJMS-1101427

We thank reviewers for their feedback, which has helped inform the quality and impact of this revised manuscript.

Reviewer 2

Reviewer comments

Changes and Amendments

The manuscript presents an interesting and useful set of experimental data on the effects of exopolysaccharides and endopolysaccharides on zebrafish embryos. However, the data on survival rate (Fig.1 ) and hatching rate (Fig. 3) of zebrafish embryos require further statistical analysis. It does not follow from the text whether the authors have adjusted the p-values for multiple comparisons. Each set of data represents 7 pairwise tests for the difference between each treatment with control. All the p-values should be corrected using Bonferroni corrected, i.e. each p-value should be multiple on 7. When it is done, the p-values for the groups exposed to low concentration of the tested chemicals should be non-significant.

Thank you for your comments and we have corrected the technical error in Figure 1 and Figure 3: (all symbols have been amended)

Figure 1 EPS: (0.1, 0.05, and 0.01 mg/mL) are not significant.

Figure 1 ENS: (0.1, 0.05, and 0.01 mg/mL) are not significant.

Figure 3 EPS: (0.5, 0.1, 0.05, and 0.01 mg/mL) are not significant.

Figure 3 EPS: (0.1, 0.05, and 0.01 mg/mL) are not significant.

One-way analysis of variance (ANOVA) was used to carry out the significant differences with a post hoc test using Dunnett’s Multiple Comparison. The significant difference was considered at *PË‚0.05, **P<0.01 and ***P<0.001 between the means of treated group as compared to zebrafish embryos in embryo media only (untreated).

Round 2

Reviewer 1 Report

Dear Authors, thank you for quickly answering. The paper has been significantly improved by abstract correction and all the questions asked have been clarified.